# The Potential Use of *Pseudomonas stutzeri* as a Biocatalyst for the Removal of Heavy Metals and the Generation of Bioelectricity

Rojas-Flores Segundo [1,*], Magaly De La Cruz-Noriega [1], Luis Cabanillas-Chirinos [2], Nélida Milly Otiniano [1], Nancy Soto-Deza [1], Walter Rojas-Villacorta [2] and Mayra De La Cruz-Cerquin [1]

[1] Institutos y Centros de Investigación de la Universidad Cesar Vallejo, Universidad Cesar Vallejo, Trujillo 13001, Peru; mdelacruzn@ucv.edu.pe (M.D.L.C.-N.); notiniano@ucv.edu.pe (N.M.O.); nsoto@ucv.edu.pe (N.S.-D.); mdelacruz@ucv.edu.pe (M.D.L.C.-C.)

[2] Investigación Formativa e Integridad Científica, Universidad César Vallejo, Trujillo 13001, Peru; lcabanillas@ucv.edu.pe (L.C.-C.); wrojasv@ucv.edu.pe (W.R.-V.)

[*] Correspondence: segundo.rojas.89@gmail.com

**Abstract:** Currently, industry in all its forms is vital for the human population because it provides the services and goods necessary to live. However, this process also pollutes soils and rivers. This research provides an environmentally friendly solution for the generation of electrical energy and the bioremediation of heavy metals such as arsenic, iron, and copper present in river waters used to irrigate farmers' crops. This research used single-chamber microbial fuel cells with activated carbon and zinc electrodes as anodes and cathodes, respectively, and farmers' irrigation water contaminated with mining waste as substrate. *Pseudomonas stutzeri* was used as a biocatalyst due to its ability to proliferate at temperatures between 4 and 44 °C—at which the waters that feed irrigated rivers pass on their way to the sea—managing to generate peaks of electric current and voltage of 4.35 mA and 0.91 V on the sixth day, which operated with an electrical conductivity of 222 mS/cm and a pH of 6.74. Likewise, the parameters of nitrogen, total organic carbon, carbon lost on the ignition, dissolved organic carbon, and chemical oxygen demand were reduced by 51.19%, 79.92%, 64.95%, 79.89%, 79.93%, and 86.46%. At the same time, iron, copper, and arsenic values decreased by 84.625, 14.533, and 90.831%, respectively. The internal resistance values shown were $26.355 \pm 4.528\ \Omega$ with a power density of 422.054 mW/cm$^2$ with a current density of 5.766 A/cm$^2$. This research gives society, governments, and private companies an economical and easily scalable prototype capable of simultaneously generating electrical energy and removing heavy metals.

**Keywords:** single-chamber microbial fuel cells; microorganisms; biocatalyst; iron; copper; arsenic

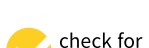

**Correction Statement:** This article has been republished with a minor change. The change does not affect the scientific content of the article and further details are available within the backmatter of the website version of this article.



## 1. Introduction

Different metals are vital for technological, agricultural, and industrial development; these areas have been developing rapidly for many years [1]. After excessive use, these metals are thrown into industrial effluents, becoming a severe problem for society [2,3]. In Peru, in recent decades, a problem has arisen in the mining sector: illegal mining has taken over the high areas of the Andes. This type of trade disposes of the products used for the extraction and purification of the metals extracted (lead iron, gold, silver, and copper) from nearby rivers [4,5]. It has been reported that almost 28% of the mining carried out in Peru is illegal, with an estimated production of 35 tons of illegally extracted gold, raising approximately USD 1 billion [6]. Furthermore, it is predicted that this activity will increase due to the lack of regulations by the Peruvian government and the increase in gold in the international market, causing the waters of the rivers used for agriculture to be contaminated, and large percentages of toxic metal ions have been found in fruits and vegetables [7,8]. Galagarza et al. (2021) carried out a study on cases with the more significant contamination of people due to the consumption of contaminated food, showing

that the Cerro de Pasco region has 3.237 infected people, which is precisely the region where there is a greater concentration of mining centers [9].

The use of methods or technologies for the purification of these waters is of vital importance. In the last decade, microbial fuel cells (MFCs) have been investigated, which have been positioned as a promising technology, not only for the generation of electrical energy, but by modifying their electrodes, it is possible to reduce the concentration of some toxic ions metals present in the substrates used [10–12]. Wang et al. (2023) manufactured MFCs with carbon/steel electrodes, managing to generate peaks of 44 mV with a power density of 14.29 mW/m$^2$, and they also managed to reduce the concentration of Cu (II) and Cr (VI) by 98.09% in 208 h [13]. Ahmad et al. (2023) used single-chamber MFCs with a graphite electrode, generating a power density of 0.69 mW/m$^2$ and decreasing the concentration of Cd$^{2+}$, Cr$^{3+}$, Pb$^{2+}$, and Ni$^{2+}$ by 81.20% [14]. Yaqoob et al. (2022) used graphene oxide and polymer–metal oxide electrodes in their microbial fuel cells, managing to generate power density peaks of 2.09 mW/m$^2$ and decreasing 78.10% and 80.25% of Pb (II) and Cd (II) in 17 days [15]. Currently, catalysts are being used as future components to improve or accelerate the decrease in heavy metal concentration [16]. Within these catalysts, there are some microorganisms (considered biocatalysts). Some species reported as biocatalysts are *Cyanothece*, *Chlorella*, *Phormidium*, *Scenedesmus*, *Chlamydomonas*, *Desmodesmus*, *Arthrospira*, *Nostoc*, and *Pseudomonas stutzeri* [17,18]. *Pseudomonas stutzeri* was investigated by Palanivel et al. (2020) to observe its potential to reduce copper in soils, managing to keep a significant decrease in this metal by approximately 85% [19]. Likewise, Ridene et al. (2023) investigated the potential of *Pseudomonas stutzeri* as a bioremediation substance for lead reduction, achieving a 71.02% reduction in soils [20]. Imron et al. (2021) used *Pseudomonas stutzeri* as a biocatalyst for the reduction of Hg, managing to reduce the concentration of this metal by 54% in 72 h [21]. The great potential that *Pseudomonas stutzeri* has as a microorganism to reduce some toxic ions has already been reported, but the compatibility of this microorganism with the simultaneous generation of electrical energy and reductions in toxic ions in microbial fuel cells is still unknown.

The main objective of this research was to observe the potential that *Pseudomonas stutzeri* has in microbial fuel cells to reduce As, Cu, and Fe and generate bioelectricity at the same time, using contaminated water from the Moche River, Trujillo, Peru as a substrate (for which the values of electric current, voltage, electrical resistance, potential density as a function of current density, electrical conductivity, nitrogen, total organic carbon, total inorganic carbon, carbon lost by ignition, dissolved organic carbon (DOC) and chemical oxygen demand (COD)will be measured). This research will result in an innovative prototype capable of reducing toxic metals for society and the water economically used for agriculture, making it a technology that is easily scalable in the short term. At present, this type of microorganism has not been investigated as a biocatalyst in microbial fuel cells for reducing toxic ions in agricultural waters, nor has its ability to be a microorganism compatible with MFC electrodes for generating electrical energy.

## 2. Materials and Methods

### 2.1. Construction of the MFC-SC

The single-chamber microbial fuel cell was manufactured from acrylic plates (Polymethylmethacrylate) of 50 × 50 cm$^2$, in which a circular hole with a 15 cm radius was made where the cathode electrode (Zinc, Zn) was placed, and the anode electrode (carbon/copper) was placed inside. The anode electrode was manufactured in the same way as that carried out by Agüero et al. (2023) [22]. Nafion 117 (Wilmington, DE, USA) was used as a proton exchange membrane to separate the anodic and cathodic chambers, as seen in Figure 1.

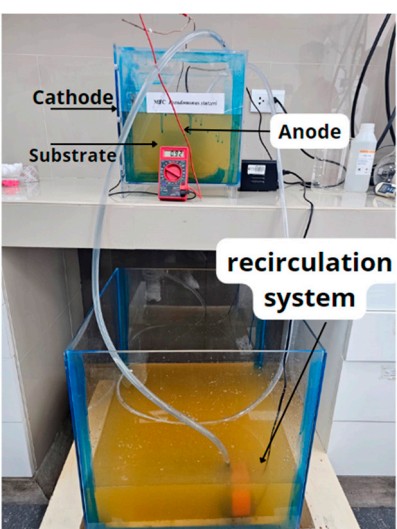

**Figure 1.** Schematization of the experimental design of the MFC-SC.

*2.2. Collection of Samples Used as a Substrate*

The substrate used was collected from the water used for irrigation from the Moche River, Trujillo, La Libertad, Peru; 120 L of this water contaminated by mining in the mountain area of La Libertad was collected. This water was gathered in drums of 25 L each and then taken to the laboratory until their later use. In addition, it can be seen that the water from these areas was used to irrigate fruits and vegetables.

*2.3. Characterization of Contaminated Waters*

The water used as substrate was characterized based on the parameters of electrical conductivity (μs/cm), pH, turbidity (NTU), nitrogen, total organic carbon (mg/L), carbon lost on ignition (mg/L), carbon dissolved organic (mg/L), and chemical oxygen demand (mg $O_2$/L). To measure the values of electrical conductivity, pH, and turbidity, multiparameter HI98194 and a TU-2016 Digital Turbidimeter were used, respectively, while for the measurement of the values of nitrogen, total organic carbon, carbon lost on ignition, dissolved organic carbon, and chemical oxygen demand, they were carried out in an external laboratory under the Standard Methods for the examination of Water and Wastewater, 23th Ed. 2017 (Part 4500. Titulometric) [22]. For the measurement of nitrogen and the other parameters (total organic carbon, carbon lost by ignition, dissolved organic carbon), the measurements were carried out under the Official Mexican Standard NOM-021-RECNAT-2000 As 07, and the measurement of chemical oxygen demand was carried out under the Standard Methods for the examination of Water and Wastewater. 23th Ed. 2017 (Part 5220-B. Open reflux). Metal atomic absorption spectrometry was used to measure the concentration of As, Cu, and Fe, following EPA Method 200.7, Rev. 4.4. The determination of the removal percentage was carried out using Equation (1), used by Lopez et al. (2022) [23].

$$Removal\ (\%) = \frac{\text{Initial concentration} - \text{Final concentration}}{\text{Initial concentration}} \times 100 \tag{1}$$

*2.4. Characterization of the Electrochemical Parameters of the MFC-SC*

The voltage and electrical current values were monitored for 12 days using a digital multimeter (Truper MUT—830 Digital Multimeter, Hialeah Gardens, FL, USA) and an external resistance of 100 Ω. The internal resistance values of the MFCs were determined using an energy sensor (Vernier- ±30 V and ±1000 mA). The power density and current density values were determined using the method of Rojas et al. (2023) using the external resistances of 0.3 (±0.1), 3 (±0.6), 10 (±1.3), 50 (±8.7), 100 (±9.3), 220 (±13), 460 (±23.1), 531 (±26.8), 700 (±40.5), and 1000 (±50.6) Ω [24].

### 2.5. Bacterial Cultures of the Electrogenic Strains

The pure culture of the *Pseudomonas stutzeri* bacteria was obtained from the Institute of Science and Technology of the Cesar Vallejo University (Trujillo, Peru). This proteobacterium was previously identified using molecular biology techniques in the work by Rojas-Villacorta et al. (2023) with an identity percentage of 100.00% [25]. (Table 1).

**Table 1.** Species identified from the anode of MFCs with pepper residues.

| Coding | Identified Species | Category | pb | % Identity | Accession Number |
|--------|-------------------|----------|------|------------|------------------|
| BAC4 | *Pseudomonas stutzeri* | Bacteria | 1439 | 100.00 | MT027239.1 |

### 2.6. Reactivation of the Pseudomonas stutzeri Strain

From an axenic culture, a suspension was made in BHI (Brain Heart Infusion) broth and incubated at 35 °C; then, it was seeded using the Striating Technique in a Petri dish with Nutrient Agar and incubated at 35 °C for 24 h. After the incubation time, a characteristic colony was selected, and Gram staining was performed to verify the purity of the bacteria [26].

### 2.7. Preparation of Bacterial Inoculum

The axenic culture of *Pseudomonas stutzeri* was sown in Petri dishes with Nutrient Agar and incubated at 37 °C for 24 h. After the incubation time, a bacterial suspension of $3 \times 10^8$ cells/mL (equivalent to Mac Farland tube No. 2) was made, and then, 10 mL of this suspension was put in a flask with 90 mL of sterile Trypticase Soy Broth; this process was carried out in triplicate or quintuplicate. The inoculated flasks were incubated at 37 °C for 24 h and shaken at 120 rpm so that growth was homogeneous and film formation was avoided. Subsequently, the absorbance was measured at 660 nm before adding them to the microbial combustion cells.

### 2.8. Commissioning of the Treatment Modules

In a container that had the function of storage or a reservoir, 120 L of water to be treated was placed in constant recirculation. The microbial combustion MFC was placed on one side; in turn, with the help of water pumps, the water was recirculated until an equilibrium flow rate of approximately 2 to 5 L/s was achieved. The electrode was placed inside the MFC. At this point, the first measurements of current, voltage, pH, and temperature, among others, were made. A visual evolution of between 60 and 90 min was considered to demonstrate the correct functioning of the treatment module before adding the bacterial inoculum (to observe that there were no leaks, obstructions of the flow rate, etc.).

## 3. Results and Analysis

Figure 2a shows the voltage values obtained from the MFC-SC during the 12 days of monitoring, observing an increase in the values from the first day in the MFC-SC of the Target (0.08 V) and cell with *P. stutzeri* (0.12 V) until the sixth, where they reached their maximum values of 0.35 and 0.91 V, respectively, and then observed a decrease in values until the last day (0.17 and 0.15 V). The literature has reported that the increase in voltage values in the first few days is due to the oxidation–reduction process between the chambers due to a large amount of organic matter [27,28]. In contrast, the decrease in potential values is inevitable because the metabolism of the microorganisms is obtained through the oxidation of the substrate used, which is depleted [29,30]. Figure 2b shows that the electrical current values increased from day 1, both for the MFC used with Target (0.11 mA) and the MFC-*Pseudomonas stutzeri* (0.13 mA) to the eighth (1.19 mA) and sixth day (4.35 mA), respectively; after these days, the values decreased until the last day. It has been reported that the increase in carbon sources in the substrate favors the growth of biofilms formed on the anode electrode, helping to generate electric current [31]. Gustave et al.

(2018) managed to create electrical current peaks of 0.31 mA using rice soil as a substrate and with a bacterial community (*Acidobacteria*, *Proteobacteria*, *Firmicutes*, *Bacteroidetes*, and *Chloroflexi*), also managing to reduce the arsenic concentration by 65% in 50 days. [32]. It has been possible to generate electric current peaks of 1.15 mA on the thirteenth day using MFC-SC with carbon electrodes while decreasing the concentration of As and Fe on the tenth day [33].

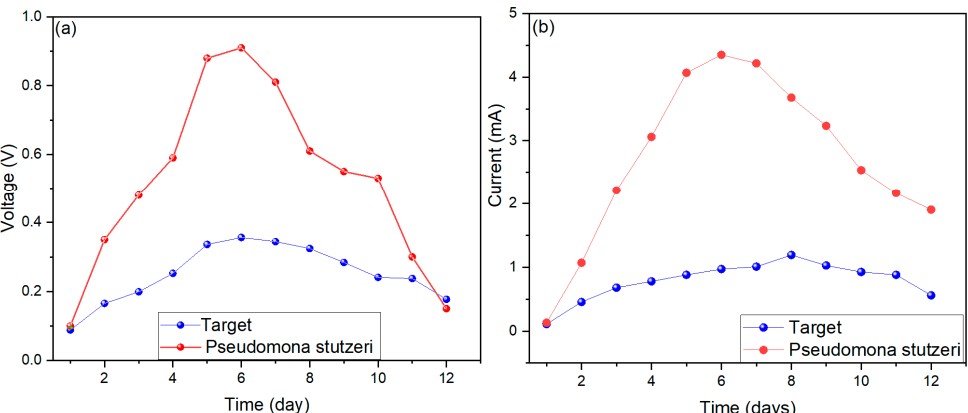

**Figure 2.** Values of (**a**) voltage and (**b**) electric current obtained from the MFCs-SC.

In Table 2, you can see the decrease in the chemical parameters measured on days 1 to 12, observing that nitrogen decreased by 66.67% (0.56 mg/L) for the MFC used as Target and 51.19% (0.82 mg/L) for the MFC with *Pseudomonas stutzeri*, compared to its initial value (1.68 mg/L). In the literature, it has been shown that the decrease in pH values stimulates the reduction in nitrogen values [34]; in the work carried out by Mandal and Das (2018), they mention the indirect electrochemical oxidation that occurs in the MFC. It is hydrogen peroxide that is responsible for this biodegradation [35]. Meanwhile, the total organic carbon values started with a value of 383.1 mg/L, managing to decrease by 79.92% (76.9 mg/L) and 71.07% (110.8 mg/L) for the MFC used as Target and with *Pseudomonas stutzeri*, respectively. The total inorganic carbon (TIC) values decreased compared to their initial value (988.0 mg/L), achieving a reduction of 64.95% (346.2 mg/L) and 60.14% (393.8 mg/L) for the MFC–Target and MFC–*Pseudomonas stutzeri*. Previous work has shown that the decrease in the percentage of TIC is because it is used as an energy source for metabolizing microorganisms [36]. In the same sense, the redox reactions within the MFC convert $CO_2$ into carbohydrate sources, allowing the organisms present in the substrate to use dissolved inorganic carbon as rich carbon sources [37]. In Table 2, it can also be seen that values of carbon lost by ignition shown on the first day (114.9 mg/L) decreased by 79.89% (23.1 mg/L) and 62.40% (43.2 mg/L) in the MFC–Target and MFC–*Pseudomonas stutzeri*, respectively. These results demonstrate that the anode can improve the performance of the bacterial oxidation of organic compounds without the need to use a biocatalyst. Still, it has been found in the literature that the location of the electrodes in the MFC can improve the performance for elimination of carbon lost by initiation depending on the position of the electrode to the cell floor [38,39]. The values of dissolved organic carbon (DOC) decreased by 79.93% (49.2 mg/L) and 65.37% (84.9 mg/L) compared to their initial values (43.2 mg/L) in the MFC–Target and MFC–*Pseudomonas stutzeri*, respectively. The decrease in these values was due to the bioelectrochemical process that originates in the MFC, promoting the leaching of the organic matter present [40]; for example, Khan et al. (2020) mentions in their research that low-molecular-weight compounds, amino acids, sugar compounds, aliphatic organic compounds (aldehydes, ketones, and ethers) and carboxylic acids are readily biodegradable and have a more significant presence in river water used for agriculture [41]. At the same time, the values of the chemical oxygen demand (COD) were observed to be lower than their initial values (416.1 mgO_2/L) by 86.46% (56.3 mgO_2/L) and 76.83% (96.4 mgO_2/L) for the MFC–Target and MFC–*Pseudomonas stutzeri*. This high

COD elimination rate is related to the high values of electric currents generated because, according to Aleid et al. (2023), high electron production gradually reduces COD values [42]. It has also been shown that in this type of water, a great variety of microorganisms use the compounds present for their metabolism as carbon sources and that the elimination of COD and generation of current are compatible [43].

**Table 2.** Initial and final chemical parameters of the MFC-SC.

| | Initial | | Final |
|---|---|---|---|
| Parameters | | Target | *Pseudomonas stutzeri* |
| Nitrogen (mg/L) | 1.68 | 0.56 | 0.82 |
| Total organic carbon (mg/L) | 383.1 | 76.9 | 110.8 |
| Total inorganic carbon (mg/L) | 988.0 | 346.2 | 393.8 |
| Carbon lost on ignition (mg/L) | 114.9 | 23.1 | 43.2 |
| Dissolved organic carbon—DOC (mg/L) | 245.2 | 49.2 | 84.9 |
| Chemical oxygen demand, COD (mgO$_2$/L) | 416.1 | 56.3 | 96.4 |

Table 3 shows the concentration values of the metals analyzed in the bioremediation process. It can be seen that the MFC–Target managed to reduce the concentration of As, Cu, and Fe by 90.83, 14.53, and 84.625%, while the MFC-*P. stutzeri* reduced As, Cu, and Fe by 90.83, 25.377, and 89.73%, respectively, in 72 h. The decrease in the values of these metals may be due to a series of natural phenomena that occur within the MFC, for example, the adhesion of these metals to the anode electrode (activated carbon) due to the porosity that this electrode presents in addition to the same metabolic activity of microorganisms that can convert these metals into non-toxic compounds [44,45]. It has also been observed in previous works that the increase in arsenic concentrations significantly decreases the energy values, because it results in a toxic substance for the electrogenic microorganisms in the anodic chambers [46], but in this research, we worked with values found in the studied river. The decrease in energy values can also be explained because the toxic arsenic, iron, and copper attached to the electrode had enough time to diffuse into the biofilm, directly affecting the cells and inhibiting the activity of bacteria [47,48]. One of the critical factors is the recovery time of the MFCs as a biosensor. The MFCs had a recovery time of approximately 72 h, where they showed a decrease in voltage values; after that time, the values increased. Until reflux was carried out in the system, the values decreased again because the system was loaded again with concentrations of arsenic, iron, and copper. However, after 72 more hours, the voltage and current values increased, thus showing a regenerative system.

**Table 3.** Heavy metals absorbed in the bioremediation process of MFCs.

| | Initial | Target | | | *P. stutzeri* | |
|---|---|---|---|---|---|---|
| | 0 h | 24 h | 72 h | 24 h | 72 h | |
| As (mg/Kg) | 0.360 | 0.210 | <0.0033 | 0.195 | <0.0033 | |
| Cu (mg/Kg) | 2.12 | 2.011 | 1.812 | 1.924 | 1.582 | |
| Fe (mg/Kg) | 15.09 | 6.374 | 2.32 | 4.671 | 1.55 | |

The pH values are shown in Figure 3a, where it can be seen that the MFC–Target values remained in the moderately acidic scale, showing a slight increase. While the MFC–*Pseudomonas stutzeri* showed values from slightly acidic to neutral, the optimal operating values demonstrated by the MFCs used as Target and *Pseudomonas stutzeri* were 4.62 and 6.74, respectively. It has been shown that variations in pH values are due to the production of protons; these variations in this parameter affect the performance and stability of MFCs [49,50]. Scientists have pointed out that the cathodes or biocathodes used

are the primary acceptors of e⁻ (electrons) because they directly capture the e⁻ coming from the external circuit. The same circuit originates from the anode electrode [51]. Velez et al. (2020) in their research managed to reduce As, Cd, Cr, Cu, Ni, Zn, Pb, and Hg, working at an acidic pH, mentioning that when the pH value decreased, it modified the oxidizing environment within an MFC, with these conditions. It reduced sulfates, nitrates, and other heavy metals more significantly [52]. Figure 3b shows the electrical conductivity values of the MFC, observing that the values increased from the first day to the sixth day for both cases, with the peak values being 109 and 222 mS/cm for the MFC used as Target and with *Pseudomonas stutzeri*, and then, a slight decrease was observed until the last day. Zhang et al. (2020) mentioned in their investigation of microbial fuel cells using irrigation waters that the main reason for the reduction of ionic heavy metals was due to the ionic copper content and the high electrical conductivity and high recorded values of electrical potential (peak values of 500 mV) in their experiment [53]. Yaqoob et al. (2020) reported in their research that carbon electrodes are an efficient material but that in the initial iteration, their electrical conductivity values were low, which is why microorganisms managed to colonize small parts of their electrode [54]. Figure 3c shows the decrease in the turbidity values of the waters used as substrate in each MFC, observing a decrease in this parameter from the first day to the last and managing to observe a reduction of 26.59 and 67.5% for the MFC used as Target and with *Pseudomonas stutzeri*, respectively. The suspended solids in the water used as substrate presented considerable values. This pattern is repeated in reported research where they mention that the decrease in these values is due to their retention time and biological degradation [55,56]. This is why the MFC with *Pseudomonas stutzeri* decreased significantly more than the MFC used as the Target. Mkilima et al. (2024) stated that high levels of turbidity are due to the dirt, mud, and particulate matter that this water drags in its process of reaching the sea because many people use it as collection centers for their organic waste [57].

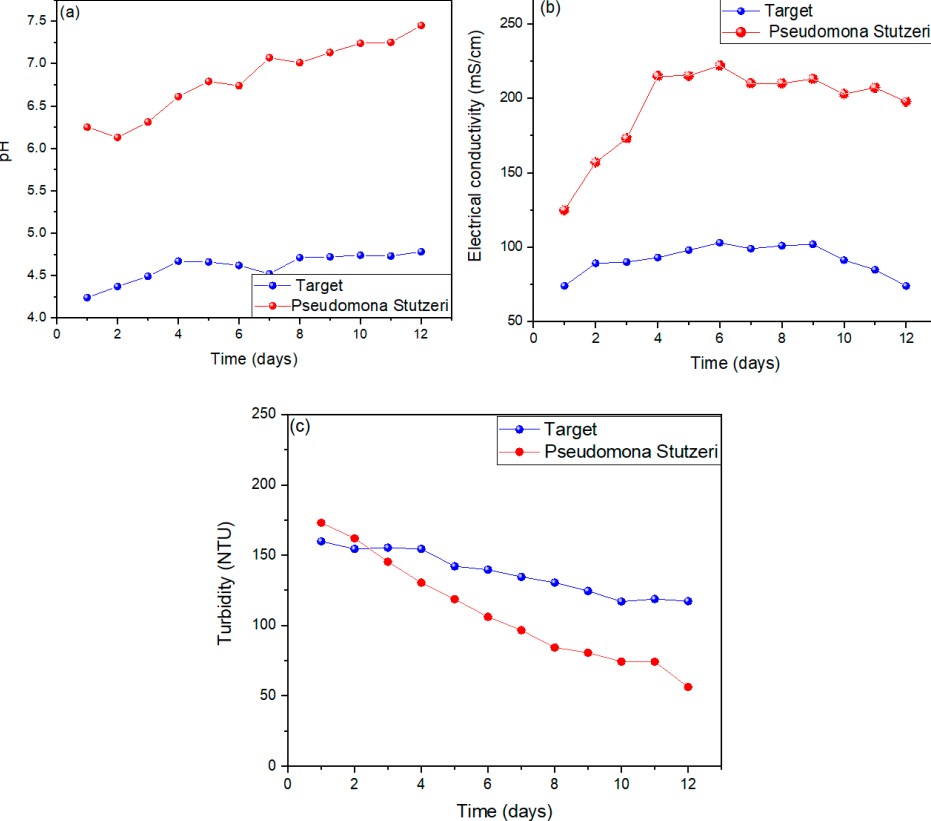

**Figure 3.** Values obtained for (**a**) pH, (**b**) electrical conductivity, and (**c**) turbidity of the MFCs substrate.

Figure 4 shows the rint values (internal resistance) of the MFCs, which were calculated using Ohm's Law (V = RI), for which the "*x*" axis was adjusted to the electric current values and the "*y*" axis was adjusted to the voltage values; performing a linear fit, the slope represents the internal resistance of the MFC [58]. The rint values for the MFC–Target and MFC–*Pseudomonas stutzeri* were 89.841 ± 11.254 and 26.355 ± 4.528 Ω, respectively. The dependence of the value of the internal resistance of electronic devices is vital because it affects the start-up time, which is why the growth of microorganisms in biofilms occurs at a higher speed at high resistance [59,60]. Yaqoob et al. (2023) used rotten rice as a substrate in their MFC, showing an internal resistance of 312.5 Ω using graphite as electrodes. They also mentioned that the increase in internal resistance decreased the mobility of electrons [61]. Likewise, Raychaudhuri et al. (2023), in their MFCs, used rice mill wastewater as a substrate, achieving an internal resistance of 372.34 Ω using carbon electrodes, mentioning that high absorbed values of the chemical oxygen demand were due to the high porosity shown of the electrodes [62]. In the literature, it has been found that metallic electrodes or electrodes with metallic inlays improve the performance of an MFC, reducing its internal resistance values due to the inherent properties that these materials have, facilitating the transport of electrons between the anodic and cathodic chambers [63]. Figure 5 shows the values of the power density (PD) as a function of the current density (CD) of the MFCs. The maximum power density found for each MFC was 309.684 and 422.054 mW/cm$^2$ in a current density of 4.756 and 5.766 A/cm$^2$ with a peak voltage of 333.418 and 891.251 mV for the MFC–Target and MFC–*Pseudomonas stutzeri*, respectively. Liu et al. (2020) showed a maximum power density of 0.198 mW/m$^2$ in their single-chamber MFCs using industrial wastewater as a substrate, mentioning that these high values are due to Pt fouling in their carbon electrodes, which were used as a catalyst [64]. Likewise, Choudhury et al. (2021) used dairy wastewater waste in their single-chamber fuel cells with Pt. They activated carbon electrodes, managing to generate power density peaks of 50 mW/m$^2$, demonstrating that the current density is primarily due to measured internal resistance values used because they influence the flow of electrons that flow through the circuit [65]. Velez et al. (2020) in their research used industrial acids from mines and municipal wastewater as a substrate in single-chamber MFCs with graphite electrodes, managing to generate power density peaks of 419 mW/m$^2$, mentioning that pathogenic microorganisms and organic and inorganic compounds present in the substrate would contribute to the generation of electrons and the efficiencies of the electronic device (microbial fuel cells) [52].

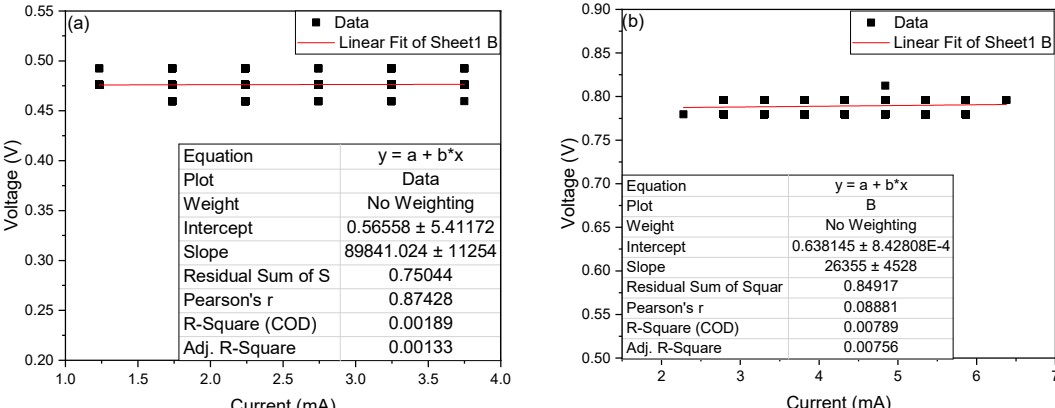

**Figure 4.** Internal resistance values of (**a**) MFC–Target and (**b**) MFC–*Pseudomonas stutzeri*.

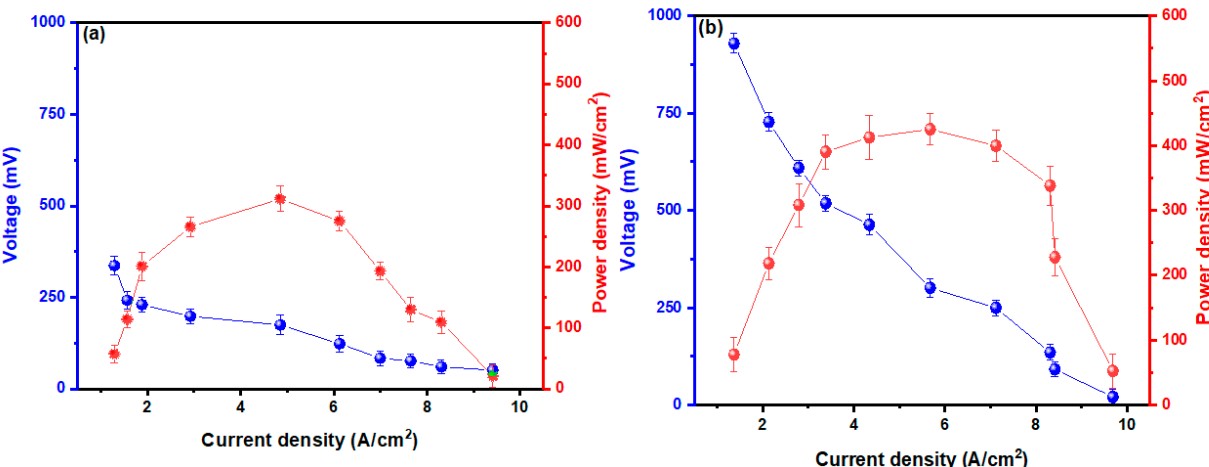

**Figure 5.** Values of the power density as a function of the current density of the (**a**) MFC–Target and (**b**) MFC–*Pseudomonas stutzeri*.

The reduction of various heavy metals has been investigated using microbial fuel cells of different designs and electrodes of different materials; some of the most recent ones can be seen in Table 4. One of the most important aspects is the reduction in costs to solve two environmental problems (eliminating heavy metals in water and generating bioelectricity). The degradation of metal ions toxic to living beings using sugar-based substrates is one of the novel techniques in this field; for example, Omenesa et al. (2023) managed to reduce the concentration of $Pb^{2+}$ by 89% using single-chamber microbial fuel cells with graphite electrodes in 12 days, where they used wastewater as a substrate, to which 50 g of sugar was added in 500 mL of substrate; they managed to generate peaks of 150 mV with a power density of 0.108 mW/m$^2$ [66]. Alshammari et al. (2024) in their research managed to generate peaks of 610 mV with a power density of 3.164 mW/m$^2$ and managed to reduce different types of heavy metals, with $Ni^{2+}$ being the one that decreased the most (95.99%); they also mentioned that the MFC technology will be easily adapted because its process is biologically stable by parameterizing chemical values. Studies have not yet widely addressed electrodes/substrates for compression from the electrochemical, biochemical, and microbial points of view [67]. Munoz C. and Bassi. A. (2024) studied an MFC with synthetic wastewater, managing to reduce Cu (II), Mg (II), Mn (II), Zn (II), and Na by 93 85, 93, 88, and 36%, respectively, and mentioned that bacterial nanowires are also formed in toxic environments and are responsible for the elimination of toxic ions present in the substrate and, at the same time, allow a higher rate of electron transport, generating greater efficiency in the MFC [68]. Other metal ions (lead, cadmium, chromium, and copper) have also been reduced using MFCs, highlighting the importance of the physiological activity of biofilms and extracellular polymeric substances for the reduction of these metals, where the use of nanoparticles and anodic and cathodic biosynthesis are the most notable points [69,70]. Treatments for the improvement of electrodes in terms of porosity and electron conduction have been intensely investigated in published documents, as well as metal nanoparticles and the isolation of microorganisms for their use as biocatalysts with the aim of re-potentiating the generation of electrical energy, which are being considered with greater importance by researchers [71,72]. The results of this research show values of voltage, electric current, and power density that are higher than those shown in Table 4 because the electrodes used were manufactured with metallic materials inside and covered with carbon, which generated an electrode with high electrical conductivity, allowing the electrons generated throughout the process to be easily transferred from the anodic to the cathodic chamber. Our iron, copper, and arsenic reduction values were 84.625, 14.533, and 90.831%, which are in the ranges shown in other investigations, thus showing the potential that our MFC has using *Pseudomonas stutzeri* as a potential candidate as a biocatalyst in the generation of electrical energy and the removal of heavy metals.

**Table 4.** Comparison of heavy metal reduction through MFCs.

| Types of MFC | Metal Ions | Recovery (%) | Electrodes | Time (Days) | Ref. |
|---|---|---|---|---|---|
| Single Chamber | $Pb^{2+}$, $Cd^{2+}$ and $Hg^{2+}$ | 89, 76.45 and 89.45 | graphite rods | 12 | [66] |
| One Chamber | $Pb^{2+}$, $Cd^{2+}$, $Cr^{3+}$ and $Ni^{2+}$ | 83.67, 84.10, 84.55 and 95.99 | graphite rods | 25 | [67] |
| Dual Chamber | Cu (II), Mg (II), Mn (II), Zn (II), and Na | 98, 49, 57, 59, and 36 | Carbon felt | 4 | [68] |
| Dual Chamber | Cu (II) and Cr(VI) | 67.09 and 37.06 | Carbon felt and stainless steel | 18 | [69] |
| Single Chamber | Pb (II), Cd (II), and Cr (III) | more than 90% | graphite | 70 | [70] |
| Single Chamber | Cu | 75 | carbon cloth | 60 | [71] |
| Dual Chamber | $Cr^{6+}$ | 99.18 ± 0.1 | carbon felts | 10 | [72] |

Some of the microbes that exist in nature can form biofilms through electrical reactions with the extracellular environment or with each other; these are called "electromicrobiomes". Many of these microorganisms have been found in soils, waters, sediments, and even in environments with a high corrosive metal content, making it easy to use these microorganisms as biocatalysts in different substrates. For example, *Shewanella oneidensis* has been reported as a biocatalyst in MFCs using sediments contaminated with $Cu^{2+}$ and $Cd^{2+}$ as a substrate, managing to reduce it by 21.8 and 18.2% during the operation period; it was mentioned that metal ions harm the operation of the cells because the electrodes and microbes reach a saturation point, limiting their life time [73]. In this same sense, Wareen et al. (2023) managed to reduce the concentrations of Zn, Ni, and Cr in their MFC by 99.8, 98.4, and 94.3% and generated voltages and PD peaks of 1120 mV and 135.14 mW/m$^2$, where they managed to identify *Leptothrix discophora* as the microorganism responsible for reducing toxic metals and generating electrical energy [74]. Other microorganisms have also been reported as biocatalysts for the reduction of heavy metals in MFCs, for example, *Klebsiella pneumoniae* [75], *Bacillus—Pseudomonas* [76], *Proteobacteria* [77], *Shewanella oneidensis* [78], and *Ochrobactrum* sp. [79]. All of these microorganisms, compared to the one used in this research (*Pseudomonas stutzeri*), did not obtain such good results in terms of voltage, electric current, and power density as those reported in Figures 1 and 5. The biofilm formed by the *Pseudomonas stutzeri* and the electrode used had excellent compatibility, making the electron transport fast and efficient; this can be seen in the values reported in Table 5 and in the results of this research.

**Table 5.** Comparison results of heavy metals bioremediation using different microbes in MFCs.

| Microbes | Metal Ions | Recovery (%) | Voltage (mV) | Power Density (mW/m$^2$) | Ref. |
|---|---|---|---|---|---|
| *Shewanella oneidensis* | $Cu^{2+}$ and $Cd^{2+}$ | 21.8 and 18.2 | 175 ± 8 | 92.43 | [73] |
| *Leptothrix discophora* | Zn, Ni and Cr | 99.8, 98.4 and 94.3 | 1120 | 135.14 | [74] |
| *Klebsiella pneumoniae* | $Cr^{3+}$, $Co^{2+}$ and $Ni^{2+}$ | 69.24, 72 and 70.11 | 102 | 99.84 | [75] |
| *Bacillus* and *Pseudomonas* | $Pb^{2+}$ | 88 | 1390 | 111.731 | [76] |
| *Proteobacteria* | $Cr^{6+}$ | 97.7 | 1020 | 55.1 | [77] |
| *Shewanella oneidensis* | Cu | 93 | 516.6 | 56.2 | [78] |
| *Ochrobactrum* sp. | $Pb^{2+}$ | 98 | 9.3 | 225.1 | [79] |

## 4. Conclusions

*Pseudomonas stutzeri* was successfully used as a biocatalyst for energy generation and bioremediation in 120 L capacity microbial fuel cells using carbon and zinc as electrodes. Values of 0.91 V and 4.35 mA were generated for the microbial fuel cell with *Pseudomonas stutzeri* on the sixth day, which operated at an optimal pH of 6.74, with an electrical conductivity of 222 mS/cm and whose turbidity was 67.5% less than its initial value. Likewise, the MFC showed a decrease in nitrogen, total organic carbon, carbon lost through ignition, dissolved organic carbon, and chemical oxygen demand of 51.19%, 79.92%, 64.95%, 79.89%, 79.93%, and 86.46%, respectively. It was also possible to reduce the arsenic, copper, and iron values in the samples by 90.83, 14.53, and 84.625%. At the same time, the peak power density values were 422.054 mW/cm$^2$ at a current density of 5.766 A/cm$^2$, with an internal resistance of 26.355 $\pm$ 4.528 $\Omega$. All these results demonstrate the viability of scaling the prototype for future work. They also demonstrate that this is an eco-friendly way to generate electrical energy and eliminate arsenic, iron, and metals, which harm agriculture and health. For future work, it is recommended to use electrodes of different sizes to increase the energy values and coat the electrodes with compounds that are not harmful to the microorganisms on the substrates.

**Author Contributions:** Conceptualization, R.-F.S.; methodology, L.C.-C.; validation, N.M.O. and N.S.-D.; formal analysis, R.-F.S and M.D.L.C.-N.; investigation, R.-F.S. and M.D.L.C.-C.; data curation, M.D.L.C.-N.; writing—original draft preparation, W.R.-V. and L.C.-C.; writing—review and editing, R.-F.S.; project administration, R.-F.S. and L.C.-C. All authors have read and agreed to the published version of the manuscript.

**Funding:** This research has been financed by the Universidad Cesar Vallejo, project code No. P-2022-125.

**Institutional Review Board Statement:** Not applicable.

**Informed Consent Statement:** Not applicable.

**Data Availability Statement:** Data is contained within the article.

**Conflicts of Interest:** The authors declare no conflicts of interest.

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
