# Peer review of "The Potential Use of Pseudomonas stutzeri as a Biocatalyst for the Removal of Heavy Metals and the Generation of Bioelectricity"

_fermentation, doi:10.3390/fermentation10020113_

Round 1

Reviewer 1 Report

Comments and Suggestions for Authors

The given manuscript "Use of Pseudomonas stutzeri as a biocatalyst in microbial fuel cells for energy generation and bioremediation" may be impactful in understanding promoting energy generation and bioremediation. Overall, the design idea of the paper is reasonable and the viewpoint is clear. However, the manuscript as it is written now, needs to be further improved. Particularly, the authors should address the following points:

1. Avoid keywords that are in the title.

2. Page 1, line 22-23: The decimal places of the values for iron, copper, and arsenic should be consistent

3. Page 2, line 56: Please write Cd2+, Cr3+, Pb2+, Ni2+ in the correct format. Please also modify other numbers in the text that require superscript or subscript.

4. Page 2, line 56-59: What are the corresponding units of 78.10 and 80.25? Change "respectively." to "respectively,".

5. Page 2, line 90: Change "Figure 01" to "Figure 1".

6. The color of the legend in Figure 3 (a) is inconsistent with the color used by the actual data line, please modify it.

7. The importance of modified electrodes for microbial fuel cell power generation and contaminant removal should be mentioned in the introduction. Several articles could be cited here, such as Bioresource Technology Volume 367, January 2023, 128230;  Applied Catalysis B: Environmental 307 (2022) 121136;  Journal of Water Process Engineering Volume 54, August 2023, 104065.

Comments on the Quality of English Language

Moderate editing of English language required

Author Response

1. Avoid keywords found in the title.
Answer. corrected title, suggested by another reviewer.

2. Page 1, line 22-23: Decimal places of iron, copper, and arsenic values must be consistent.
Answer. corrected

3. Page 2, line 56: Write Cd2+, Cr3+, Pb2+, Ni2+ in the correct format. Also modify other numbers in the text that require superscript or subscript.
Answer. corrected

4. Page 2, lines 56-59: What are the corresponding units of 78.10 and 80.25? Change "respectively". to "respectively".
Ans corrected

5. Page 2, line 90: Change "Figure 01" to "Figure 1".
Ans corrected

6. The color of the legend in Figure 3(a) does not match the color used by the actual data line; modify it.
Answer Dear colleague, it seems to be a conversion error at the time of writing the manuscript.
7. The importance of modified electrodes for microbial fuel cell power generation and contaminant removal should be mentioned in the introduction. Several articles could be cited here, such as Bioresource Technology Volume 367, January 2023, 128230;
Applied Catalysis B: Environmental 307 (2022) 121136; Journal of Water Process Engineering Volume 54, August 2023, 104065.
Ans. Some small paragraphs were placed on the importance of the electrodes and citations 11 and 12 of the recommended documents were made.

Kind regards

Reviewer 2 Report

Comments and Suggestions for Authors

This study presents a novel solution for generating electrical energy and effectively degrading heavy metals in polluted river water. Remarkable reductions in arsenic, iron, and copper were achieved. However, before considering publication, certain concerns need to be addressed.

1. In order to justify the selection of Pseudomonas stutzeri as the biocatalyst, the advantages of this strain should be highlighted in the abstract.

2. It is strongly recommended to include a table comparing the results of MCF/Heavy metal bioremediation with other reported studies, in order to emphasize the advancements made in this study.

3. Furthermore, it is highly advisable to include another table comparing the results of Heavy metal bioremediation using different microbes.

4. The full text needs to be thoroughly checked and revised, paying attention to capital letters, superscripts, subscripts, and appropriate spacing between numbers and units.

5. L138, Table 1 is not placed correctly.

6. L194, Table 1 should be renamed as Table 2; L238, Table 2 should be Table 3.

7. Tables should be formatted as three-line tables.

Author Response

Dear colleague, I hope you are feeling very well.
The authors have reviewed your suggestions in order to improve the manuscript and we have made the suggested changes in response to each of your questions.
1. To justify the selection of Pseudomonas stutzeri as a biocatalyst, the advantages of this strain must be highlighted in the summary.
Ans. got high

2. It is highly recommended to include a table comparing the MCF/heavy metal bioremediation results with other reported studies, to emphasize the progress made in this study.
Ans. Performed in Table 4

3. In addition, it is highly recommended to include another table that compares the results of bioremediation of heavy metals using different microbes
Ans. Performed in Table 5
.

4. The entire text needs to be carefully checked and revised, paying attention to capital letters, superscripts, subscripts, and appropriate spacing between numbers and units.
Ans. corrected
5. L138, Table 1 is not positioned correctly.
Answer. corrected

6. L194, Table 1 should be renamed Table 2; L238, Table 2 should be Table 3.
Answer. corrected

7. Tables should be formatted as three-line tables.
Ans corrected

Kind regards

Reviewer 3 Report

Comments and Suggestions for Authors

Reviewer comments:

The authors have presented their work in a manuscript entitled “Use of Pseudomonas stutzeri as a biocatalyst in microbial fuel cells for energy generation and bioremediation.” However, there are a lot of errors and the English language is poor in the whole Manuscript (MS). The manuscript can be accepted for publication in the Fermentation journal after major revisions
.

1.                  The novelty and significance of the current MS has not been specified and justified. The same should be reflected in the Abstract and Introduction section.

2.                  The scientific names should always be in italics.

3.                  The title must be revised to make it catchy and informative.

4.                  In the beginning, the type of paper is not specified. Is it a review or research article?

5.                  The English language needs major improvement in the whole manuscript. There are a lot of grammatical errors, and some sentences should be simplified. For Example: Line 40, 47-50, Line 53 (Who is I here?)

6.                  There is a lot of discussion of previous reports in the introduction just to add to the length of the introduction section. There is no need to add this much, and some discussions must be deleted.

7.                  Line 72: Objective of the research ‘was’.

8.                  Lines 62-63, 133, 150, 172, and many more: Scientific names should be in italics.

9.                  There is no need to add Fig. 2. It can be omitted.

10.              Journal guidelines have not been followed for the citations in the MS.

11.              Table 1, 2: What does the ‘Target’ represent here?

12.              Line 200-203: revise this line with respect to the English language. The english language is poor throughout the Manuscript.

13.              Line 211: CO2

14.              The authors have not highlighted the significance of their research work anywhere in the manuscript.

15.              The results are not well discussed and compared with the recent studies. The results of the present study have not been properly compared with the reported studies from the literature. The manuscript lacks a proper discussion and interpretation of the data.

Comments on the Quality of English Language

English language is poor in the MS and needs major improvement. 

Author Response

Dear colleague, I hope you are feeling very well.
The authors have reviewed your suggestions in order to improve the manuscript and we have made the suggested changes in response to each of your questions.
1. The novelty and significance of the current EM has not been specified or justified. The same should be reflected in the Summary and Introduction section.
Ans. In the summary it was placed in the last part of it, just as its importance was also placed in the last part of the introduction.
Line 71-73 and 82-87
2. Scientific names must always be in italics.
Answer. corrected.
3. The title should be revised to make it attractive and informative.
Answer. corrected.
4. At the beginning the type of paper is not specified. Is it a review or research article?
Answer. Research, was corrected
5. The English language needs major improvements throughout the manuscript. There are many grammatical errors and some sentences should be simplified. For example: Line 40, 47-50, Line 53 (Who am I here?)
Ans. They were corrected
6. There is a lot of discussion of previous reports in the introduction just to increase the length of the introduction section. Not much needs to be added and some discussions need to be removed.
Ans. I understand, but the other reviewers asked me to add a few more lines.
7. Line 72: Objective of the investigation “was”.
Answer. corrected

8. Lines 62-63, 133, 150, 172 and many more: Scientific names should be italicized.
Answer. corrected

9. It is not necessary to add Fig. 2. It can be omitted.
Answer. was corrected

10. The journal's guidelines for citations in the MS have not been followed.
Answer. We authors have submitted other manuscripts to the journal, and we have always followed the same citation formats. If it is something specific, would you be so kind as to tell us. Apologies for the inconvenience.

11. Table 1, 2: What does the “objective” represent here?
Answer. Represents MFC without Pseudomona Stutzeri.

12. Line 200-203: Review this line with respect to the English language. The English language is poor throughout the Manuscript.
Answer. was corrected

13. Line 211: CO2
Answer. was corrected

14. The authors have not highlighted the importance of their research work anywhere in the manuscript.
Ans. End of the introduction.

15. The results are not well discussed or compared with recent studies. The results of the present study have not been adequately compared with studies reported in the literature. The manuscript lacks adequate discussion and interpretation of the data.
Ans. Tables 4 and 5 were placed.

Kind regards

Round 2

Reviewer 1 Report

Comments and Suggestions for Authors

The authors have well revised the MS for the given suggestion and is now acceptable for publication

Reviewer 3 Report

Comments and Suggestions for Authors

The manuscript can now be accepted for publication in the Fermentation journal.